# Response to Brigatinib Targeted Therapy in Non-Small Cell Lung Cancer Harboring Epidermal Growth Factor Receptor Exon 19 Deletion, T790M, and cis-C797S Triple Mutations: A Case Report

**DOI:** 10.3390/ijms24010602

**Published:** 2022-12-29

**Authors:** Zi-Wei Chen, Gigin Lin, Hsuan-Jen Shih, Chiao-En Wu

**Affiliations:** 1Department of Internal Medicine, Chang Gung Memorial Hospital at Linkou, Chang Gung University College of Medicine, Taoyuan 333423, Taiwan; 2Department of Medical Imaging & Intervention, Clinical Metabolomics Core, Chang Gung Memorial Hospital at Linkou, Chang Gung University College of Medicine, Taoyuan 333423, Taiwan; 3Division of Hematology-Oncology, Department of Internal Medicine, Chang Gung Memorial Hospital at Linkou, Chang Gung University College of Medicine, Taoyuan 333423, Taiwan

**Keywords:** brigatinib, NSCLC, T790M-cis-C797S, osimertinib resistance, EGFR triple mutation, *EGFR*-tyrosine kinase inhibitors resistance

## Abstract

Epidermal growth factor receptor (*EGFR*) triple mutations with exon 19 deletion (del19), T790M, and cis-C797S (del19/T790M/cis-C797S mutations) frequently occur in patients with non-small cell lung cancer (NSCLC), while progression to frontline *EGFR*-tyrosine kinase inhibitors (TKIs) and osimertinib was resistant to all clinically available *EGFR*-TKIs. Brigatinib monotherapy may be a potential treatment for NSCLC harboring del19/T790M/cis-C797S mutations based on preclinical studies; however, no clinical report has evaluated its efficacy on *EGFR* del19/T790M/cis-C797S mutations. Herein, we present a case of a female patient with *EGFR* del19-mutated NSCLC treated with afatinib followed by osimertinib due to acquired T790M mutation. The *EGFR* del19/T790M/cis-C797S mutations were detected following osimertinib treatment. Complete response of skull metastasis was confirmed after brigatinib treatment (90 mg daily). Unfortunately, she experienced intolerable adverse events; therefore, brigatinib was discontinued after three-month usage. This report provides the first reported evidence for the use of brigatinib monotherapy in patients with NSCLC harboring *EGFR* del19/T790M/cis-C797S mutations after progression to previous *EGFR*-TKIs.

## 1. Introduction

Non-small cell lung cancer (NSCLC) with epidermal growth factor receptor (*EGFR*) mutations accounts for 46.5–51.7% of NSCLC cases in Asian populations [1]. In all *EGFR* mutations, both exon 19 deletion (del19) and L858R were the majority and so called common mutation [1]. Epidermal growth factor receptor tyrosine kinase inhibitors (TKIs) have a higher response rate and longer progression-free survival (PFS) than chemotherapy in patients with *EGFR* mutations [2,3]. Therefore, *EGFR*-TKIs have been approved as the first-line treatment for patients with NSCLC harboring *EGFR* mutations. Although the benefits of molecule-targeting drugs were significant, most patients had disease progression within 1–2 years due to acquired resistance to frontline *EGFR*-TKIs, such as gefitinib, erlotinib, or afatinib. Acquired resistance was caused by a second T790M mutation, in which methionine replaced threonine at position 790. T790M blocks the ATP-binding site of *EGFR*-TKIs and *EGFR*, accounting for approximately 60% of acquired resistant cases [4,5,6,7,8,9]. 

To solve this clinical difficulty, the third-generation *EGFR*-TKI, osimertinib, which selectively targets T790M, was evaluated in late-phase NSCLC patients who acquired the *EGFR* T790M mutation. Response rate and median PFS of third-generation *EGFR*-TKIs are better than those of conventional chemotherapy [10]. However, several months after third-generation *EGFR*-TKI therapy, some patients developed *EGFR* C797S mutation, which contributes to acquired resistance to third-generation *EGFR*-TKIs in NSCLC [11]. Although the C797S mutation that emerged in trans of the T790M allele was sensitive to combined therapy with first- and third-generation *EGFR*-TKIs, the C797S mutation that emerged in cis of the T790M allele remains broadly resistant, which is a clinical obstacle that needs to be overcome; however, several studies have been conducted on this issue. Some studies have reported that brigatinib can effectively inhibit triple mutant (del19/T790M/cis-C797) *EGFR* in vitro and in vivo [12]. A preclinical study [13] also revealed excellent results of combining brigatinib and cetuximab for treating NSCLC with triple mutants containing *EGFR* C797S. As such, brigatinib alone may be an effective therapy for treating triple-mutant NSCLC, but no previous study has reported the use of brigatinib alone. Herein, we report a successful case of using brigatinib alone to overcome del19/T790M/cis-C797 mutations after progression with afatinib and osimertinib therapy.

## 2. Case Presentation

A 66-year-old Taiwanese female with no history of smoking was diagnosed in July 2017 with lower upper lobe (LUL) lung adenocarcinoma, stage IV (T2N1M1a, AJCC 7th), with malignant pleural effusion (clinical course is summarized in Figure 1). Due to detection of positive *EGFR* del19 from tumor biopsy specimens using the method of Competitive Allele-Specific TaqMan^®^ PCR (ABI) (Figure 2A), the patient began receiving 40 mg of afatinib per day for first-line treatment. The best objective response was partial response (PR), according to the Response Evaluation Criteria in Solid Tumors, version 1.1. However, 14 months after the day afatinib was initiated, the patient had progressive disease over the lung with satellite nodules in the LUL, in addition to left pleural metastasis. The *EGFR* T790M mutation was identified in a lung tissue biopsy using the Scorpions & Amplification Refractory Mutation System (QIAGEN) (Figure 2B). Based on these result, 80 mg of osimertinib daily was used as second-line treatment, and the best response to osimertinib was PR. Unfortunately, computed tomography (CT) indicated further progression of the left lower lobe pleural metastasis after 15 months of osimertinib. Therefore, hyperthermic intrathoracic chemotherapy and wedge resection of the pleural nodule were performed for local lung progression in February 2020. Subsequently, osimertinib was resumed. Nevertheless, CT revealed recurrence in the left lower lobe. Next generation sequencing (NGS, GenPan52-TBx, Genconn Biotech, Taiwan) of specimens from CT-guided biopsy of left lower lobe tumor revealed a new C797S mutation, which developed in cis with the T790M allele, in addition to the remained exon 19 deletion and T790M mutations presented before treatment with osimertinib (Figure 3A). It has been reported that sensitivity to all approved *EGFR*-TKIs was lost when C797S and T790M mutations developed in the cis form. Therefore, we decided to perform radiation therapy for the local progression of the left lower lobe tumor. Unfortunately, CT showed liver metastasis and bony metastases to C6, C7, and T2 vertebrae four months after the completion of previous radiation therapy, and the 19del/T790M/cis-C797S mutation was detected again by NGS from liver biopsy (Figure 3B). Brigatinib treatment with or without cetuximab was suggested to the patient, but she refused because of concerns regarding early onset pulmonary events from brigatinib. Therefore, palliative radiotherapy for spinal metastases and Xgeva^®^ (denosumab) for bony metastases, followed by immunochemotherapy with atezolizumab (1500 mg), bevacizumab (7.5 mg/m^2^), of pemetrexed (500 mg/m^2^), and cisplatin (75 m/m^2^), which was administered intravenously every 3 weeks in March 2021. Partial response was achieved after three months of treatment. However, after five months of immunochemotherapy, a new skull (parietal bone) metastasis was identified on brain magnetic resonance imaging. As a result, we decided to start 90 mg of brigatinib once daily to alleviate bony metastasis between November 2021 and January 2022. During the therapy period, a complete response was noted over the skull (Figure 4), and a stable status was observed over lung cancer and liver metastasis. She discontinued brigatinib in January 2022 because of intolerable myositis and elevated creatine phosphokinase (CPK).

## 3. Discussion

Herein, we reported a patient with NSCLC harboring *EGFR* 19del-T790M-cis-C797S mutation and was treated with brigatinib monotherapy as fourth-line treatment after previous treatment of afatinib, osimertinib, and immunochemotherapy.

Of all triple *EGFR* mutations in patients receiving second-line osimertinib monotherapy (such as our case), C797S is the most common mutation, accounting for about 24% [11,14]. Additionally, both C797S in cis- and trans-form with T790M may present. It was reported that most mutations of C797S were in cis form with maintained T790M rather than that in trans-form [15]. When tumors develop T790M-cis-C797S mutations, limited clinical treatments have been reported to be useful in controlling the disease. A recent clinical study [16] reported a combination therapy with brigatinib and cetuximab in five patients with lung adenocarcinoma that was positive for *EGFR* 19del-T790M-cis-C797S mutation. The combined therapy led to tumor shrinkage in three patients with an overall objective response rate of 60% and disease control rate of 100% [16]. The effectiveness of the combination of brigatinib and cetuximab inspired us to explore the clinical benefits of brigatinib alone in the treatment of lung adenocarcinoma with *EGFR* triple mutations. Thus, we reported the case of a 66-year-old woman who had lung adenocarcinoma positive for 19del/T790M/cis-797S mutation and received brigatinib (90 mg daily) for three months. Brigatinib therapy successfully controlled the lesion in the LUL and liver and decreased the size of metastatic tumors in the skull. Although we chose a relatively low dose of brigatinib compared to the recommended dosage for anaplastic lymphoma kinase-rearranged NSCLC, the adverse effects of brigatinib, such as elevated CPK and myositis, prevent patients from receiving treatment. 

Despite the good clinical response to brigatinib therapy in our case report, the reliability of the results is limited by one patient and short period of treatment. Moreover, this patient only received brigatinib for a limited period, owing to intolerable adverse effects of brigatinib; therefore, the long-term efficacy of brigatinib is unknown. However, to the best of our knowledge, this is the first report on the use of brigatinib monotherapy, which showed tumor response in patients with triple 19del/T790M/cis-797S *EGFR* mutations. 

## 4. Conclusions

Altogether, our case study demonstrated a partial response to brigatinib therapy in lung adenocarcinoma with skull metastasis harboring *EGFR* triple 19del/T790M/cis-C797S mutations, and the adverse effects caused by brigatinib still deserve more attention.

## Figures and Tables

**Figure 1 ijms-24-00602-f001:**
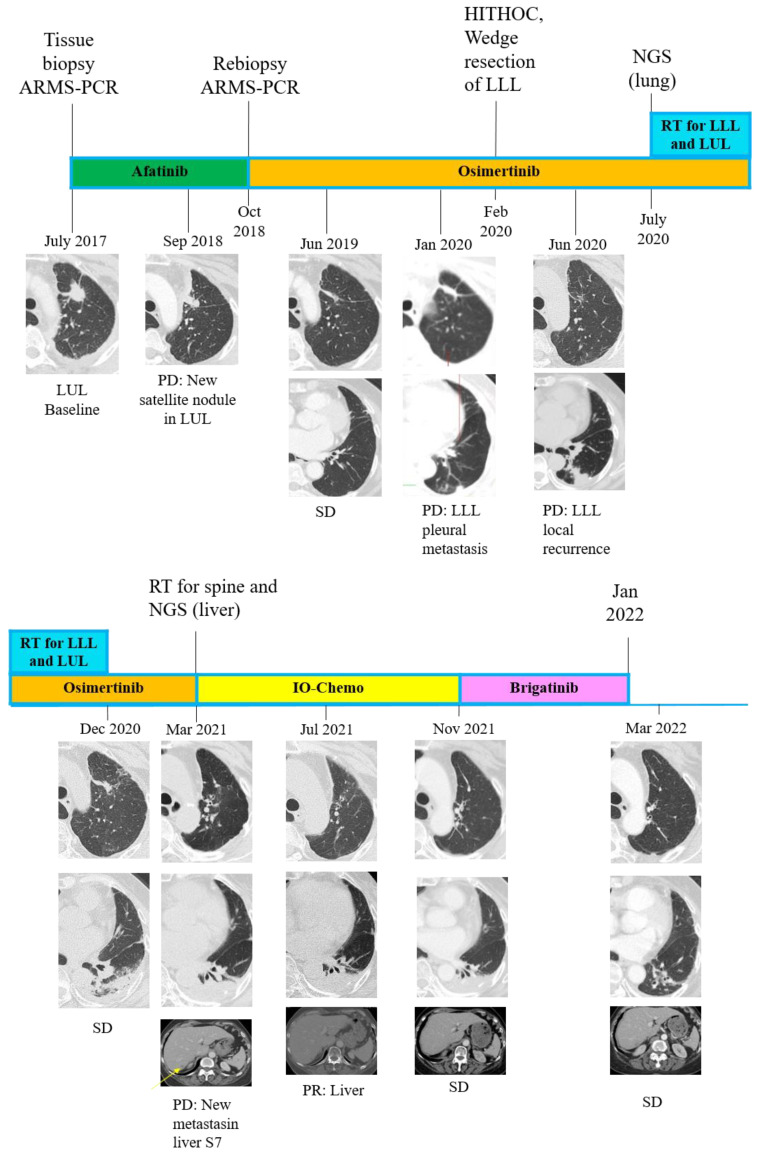
Schematics showing a timeline for the treatment course. Computed tomography (CT) showed a clinical response to epidermal growth factor receptor tyrosine kinase inhibitors (e.g., afatinib, osimertinib) and immune-oncology chemotherapy (IO-Chemo) treatment. The patient had a new satellite nodule in the left upper lobe (LUL) after 14 months of afatinib treatment. The patient had stable disease after osimertinib treatment for two months, but she eventually experienced progressive disease with pleural metastasis in the left lower lobe (LLL) after receiving osimertinib for 15 months. Thus, hyperthermic intrathoracic chemotherapy (HITHOC) and wedge resection of the pleural nodule were performed for local progression in February 2020. However, progression of LUL and LLL was found, so she underwent radiation therapy for lung lesions and remained stable for the next eight months. Subsequently, the disease metastasized to C6, C7, and T2 vertebrae and segment seven (S7) of the liver. After radiation therapy for vertebrae and four months of immunochemotherapy, regression of the liver nodule and C-spine tumor were observed. However, a new skull (parietal bone) metastasis was identified on brain, and brigatinib was then administered to treat the skull bone metastasis (Showed in Figure 4). During three months of brigatinib treatment, the lung lesion did not deteriorate, and complete response of the skull metastasis was observed. PD, progressive disease; SD, stable disease; PR, partial response; ARMS-PCR, amplification refractory mutation system–polymerase chain reaction; HITHOC, hyperthermic intrathoracic chemotherapy; NGS, next generation sequencing; RT, radiation therapy; IO-chemo, immunochemotherapy.

**Figure 2 ijms-24-00602-f002:**
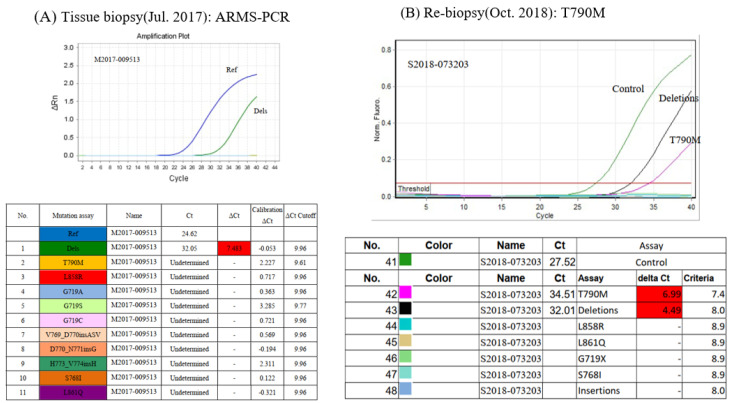
(**A**) Detection of *EGFR* exon 19 deletion in tumor tissues before treatment using Competitive Allele−Specific TaqMan^®^ PCR (ABI) in July, 2017. (**B**) Scorpions & ARMS (QI−AGEN) remained exon 19 deletion and new *EGFR* mutation T790M in tumor tissues at progression during afatinib treatment in October, 2018. ARMS−PCR, amplification refractory mutation system−polymerase chain reaction.

**Figure 3 ijms-24-00602-f003:**
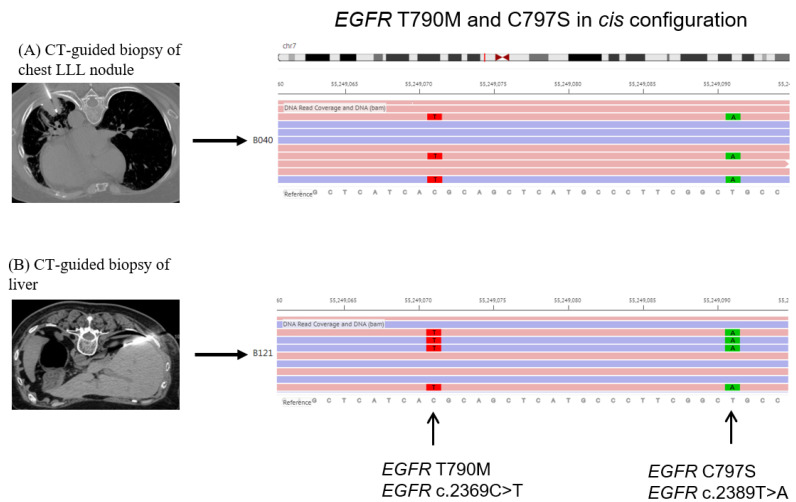
Next generation sequencing (provided by GenPan52-TBx, Genconn Biotech, Taiwan) from biopsies of LLL nodule (**A**) and liver (**B**) both showed *EGFR* T790M and C797S in cis-configuration.

**Figure 4 ijms-24-00602-f004:**
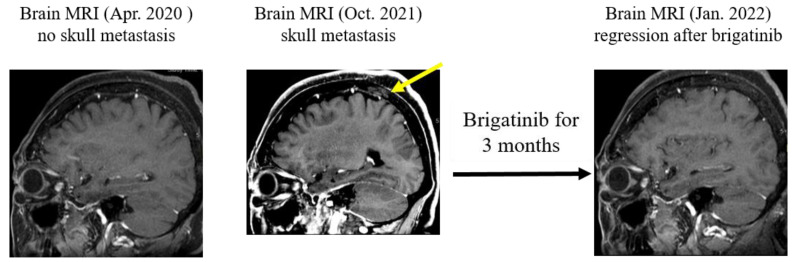
During immunochemotherapy, the patient had headache and dizziness. Brain magnetic resonance imaging (MRI) showed focal metastasis in the parietal skull (middle panel in October 2021). Three months after brigatinib treatment, the skull lesion regressed (right panel in January 2022).

## Data Availability

Data is unavailable due to privacy or ethical restrictions.

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
