# Peer review of "Response to Brigatinib Targeted Therapy in Non-Small Cell Lung Cancer Harboring Epidermal Growth Factor Receptor Exon 19 Deletion, T790M, and cis-C797S Triple Mutations: A Case Report"

_ijms, 2022, doi:10.3390/ijms24010602_

Round 1

Reviewer 1 Report

IJMS (ISSN 1422-0067)

In the article titled ‘Response to brigatinib targeted therapy in non-small cell lung cancer harbouring epidermal growth factor receptor exon 19 deletion, T790M, and cis-C797S triple mutations: A case report’ the very important issue of the emergence of resistance to TKI treatment in the presence of the T797 cis mutation relative to the original T790 resistance mutation is addressed.

The authors describe a case of a patient with adenocarcinoma of the lung treated with TKI. This is an interesting clinical case, taking into account the molecular profile of the patient and the course of sequential treatment with the last phase in the form of brigatinib.

Below are some comments on the manuscript.

1.       EGFR should be written in italics throughout the manuscript where the gene is mentioned (e.g. lines 38, 41, 43, etc.).

2.       The authors describe which qPCR kit they used to identify mutations in EGFR, but do not mention what panel was used to identify variants using NGS. Might be worth adding to it.

3.       Line 102-103: ‘…immunochemotherapy with atezolizumab, bevacizumab, pemetrexed, and cisplatin were administered.’ In what doses, and what distribution of administration?

4.       Line 114: ‘…brigatinib monotherapy.’ It could be supplemented with a statement in which line it was used.

5.       Line 115-116: r ‘…C797S contributes to acquired resistance 115 to third-generation TKIs…’ It could be added here in what percentage of patients treated with third-generation TKI this mutation is present in the cis and trans system.

6.       Line 134-136: Despite the good clinical response to brigatinib therapy in our case report, the reliability of the results is limited by the small number of patients and short period of treatment. I'm not sure if we can talk about the small number of patients here, because there was only one. I know what it's about, but it needs to be rewritten.

Author Response

Response to Reviewer 1 comments

IJMS (ISSN 1422-0067)

In the article titled ‘Response to brigatinib targeted therapy in non-small cell lung cancer harbouring epidermal growth factor receptor exon 19 deletion, T790M, and cis-C797S triple mutations: A case report’ the very important issue of the emergence of resistance to TKI treatment in the presence of the T797 cis mutation relative to the original T790 resistance mutation is addressed.

The authors describe a case of a patient with adenocarcinoma of the lung treated with TKI. This is an interesting clinical case, taking into account the molecular profile of the patient and the course of sequential treatment with the last phase in the form of brigatinib.

Below are some comments on the manuscript.

  1. EGFR should be written in italics throughout the manuscript where the gene is mentioned (e.g. lines 38, 41, 43, etc.).

Reply:

Thank for your advice. We have modified the font of EGFR in the article. (We’ve highlighted them in yellow. Please see the attachment)

  1. The authors describe which qPCR kit they used to identify mutations in EGFR, but do not mention what panel was used to identify variants using NGS. Might be worth adding to it.

Reply:
Thank for your advice. We used NGS (provided by GenPan52-TBx, Genconn Biotech, Taiwan) to identify mutations in CT-guided tumor biopsy, which information is added to the manuscript (line 91)

  1. Line 102-103: ‘…immunochemotherapy with atezolizumab, bevacizumab, pemetrexed, and cisplatin were administered.’ In what doses, and what distribution of administration?

Reply:
We use atezolizumab 1500mg, bevacizumab 7.5mg/m2, pemetrexed 500mg/m2, and cisplatin 75m/m2 (intravenous infusion, every 3 weeks), which is added in line 105, 106, 107.

  1. Line 114: ‘…brigatinib monotherapy.’ It could be supplemented with a statement in which line it was used.

Reply:
Brigatinib was used in fourth-line treatment after afatinib for first-line, osimertinib for  second-line, and immunochemotherapy for third-line treatment, which is added in lines 118,119, 120. Please see the attachment

  1. Line 115-116: r ‘…C797S contributes to acquired resistance to third-generation TKIs…’ It could be added here in what percentage of patients treated with third-generation TKI this mutation is present in the cis and trans system.

Reply:

Osimertinib can be used as the first-line or second-line therapy for EGFR mutation in non-small cell lung cancer [1]. Up to now, the T790M mutation has not been observed in patients who received first-line osimertinib treatment [2], which may be due to the highly selective inhibition of T790M by osimertinib. Therefore, the triple mutation (EGFR 19del-T790M- C797S) did not occur in the first-line osimertinib treatment, but only occurred in second-line treatment.

In patients receiving second-line osimertinib monotherapy (such as our case), triple EGFR mutation was observed in one third of patients, as a mechanism of the acquired drug resistance [3]. Of all triple EGFR mutations, C797S is the most common mutation, accounting for about 24% [1, 3]. And both C797S in cis and trans-form with T790M may present. It was reported that nearly all mutations of C797S were in cis form with maintained T790M rather than that in trans-form [4]. Moreover, a clinical case was also reported. After combining the first- and the third-generation EGFR TKIs, the trans-form C797S may transformed into cis-form C797S [5]. Therefore, C797S in cis-form with T790M is more common than that in trans-form.

Information mentioned above is summarized and added in line 121~125. Please see the attachment

  1. Line 134-136: Despite the good clinical response to brigatinib therapy in our case report, the reliability of the results is limited by the small number of patients and short period of treatment. I'm not sure if we can talk about the small number of patients here, because there was only one. I know what it's about, but it needs to be rewritten.

Reply: Thank for your advice, we have modified the grammatical error in the article (line 143). Please see the attachment

Reference

1          Leonetti A, Sharma S, Minari R, Perego P, Giovannetti E, Tiseo M. Resistance mechanisms to osimertinib in EGFR-mutated non-small cell lung cancer. British journal of cancer 2019; 121: 725-737.

2          Ramalingam S, Cheng Y, Zhou C, Ohe Y, Imamura F, Cho B et al. Mechanisms of acquired resistance to first-line osimertinib: preliminary data from the phase III FLAURA study. Annals of Oncology 2018; 29: viii740.

3          Yang Z, Yang N, Ou Q, Xiang Y, Jiang T, Wu X et al. Investigating Novel Resistance Mechanisms to Third-Generation EGFR Tyrosine Kinase Inhibitor Osimertinib in Non–Small Cell Lung Cancer PatientsNovel Resistance Mechanisms to Osimertinib in NSCLC Patients. Clinical Cancer Research 2018; 24: 3097-3107.

4          Oxnard GR, Hu Y, Mileham KF, Husain H, Costa DB, Tracy P et al. Assessment of Resistance Mechanisms and Clinical Implications in Patients With EGFR T790M-Positive Lung Cancer and Acquired Resistance to Osimertinib. JAMA Oncol 2018; 4: 1527-1534.

5          Wang Z, Yang JJ, Huang J, Ye JY, Zhang XC, Tu HY et al. Lung Adenocarcinoma Harboring EGFR T790M and In Trans C797S Responds to Combination Therapy of First- and Third-Generation EGFR TKIs and Shifts Allelic Configuration at Resistance. J Thorac Oncol 2017; 12: 1723-1727.

Reviewer 2 Report

After evaluating the manuscript "Response to brigatinib targeted therapy in non-small cell lung cancer harboring epidermal growth factor receptor exon 19 deletion, T790M, and cis-C797S triple mutations: A case report" I have to recommend its rejection in the current version.

Despite the high relevance of the problem of Non-Small-Cell Lung Cancer (NSCLC), the description of a single clinical case is not indicative.

Major

1. The list of cited literature is extremely poor and should be expanded.

2. All figures are presented in extremely low quality. They need to be revised from the point of view of clarity and the quality should be improved by at least 600 dpi.

3. The authors did not prepare an supplementary file.

Author Response

Response to Reviewer 2 comments

Comments and Suggestions for Authors :
After evaluating the manuscript "Response to brigatinib targeted therapy in non-small cell lung cancer harboring epidermal growth factor receptor exon 19 deletion, T790M, and cis-C797S triple mutations: A case report" I have to recommend its rejection in the current version.

Despite the high relevance of the problem of Non-Small-Cell Lung Cancer (NSCLC), the description of a single clinical case is not indicative.

Major

  1. The list of cited literature is extremely poor and should be expanded.---

Reply:
As the rarity of triple mutation, the relevant references are limited. We have included more reference for this case report as many as possible. Three references, numbered 14, 15 and 16, were added in this paper, please see the attachment.

14: Yang Z, Yang N, Ou Q, Xiang Y, Jiang T, Wu X et al. Investigating Novel Resistance Mechanisms to Third-Generation EGFR Tyrosine Kinase Inhibitor Osimertinib in Non–Small Cell Lung Cancer PatientsNovel Resistance Mechanisms to Osimertinib in NSCLC Patients. Clinical Cancer Research 2018; 24: 3097-3107.

15: Leonetti A, Sharma S, Minari R, Perego P, Giovannetti E, Tiseo M. Resistance mechanisms to osimertinib in EGFR-mutated non-small cell lung cancer. British journal of cancer 2019; 121: 725-737.

16: Oxnard GR, Hu Y, Mileham KF, Husain H, Costa DB, Tracy P et al. Assessment of Resistance Mechanisms and Clinical Implications in Patients With EGFR T790M-Positive Lung Cancer and Acquired Resistance to Osimertinib. JAMA Oncol 2018; 4: 1527-1534.

  1. All figures are presented in extremely low quality. They need to be revised from the point of view of clarity and the quality should be improved by at least 600 dpi.---

Reply: Thanks for your suggestion. We have adequately revised the figures to improve the quality of figures. Please see the attachment

  1. The authors did not prepare a supplementary file.

Reply: We don’t have supplementary data for this case report. And we hope that reviewer can tell us what part in the article isn't complete and needs a supplementary file.

Round 2

Reviewer 2 Report

After evaluating the manuscript “Response to brigatinib targeted therapy in non-small cell lung cancer harboring epidermal growth factor receptor exon 19 deletion, T790M, and cis-C797S triple mutations: A case report" I have to recommend its accept in current version